# Semiautomated Segmentation and Volume Measurements of Cervical Carotid High-Signal Plaques Using 3D Turbo Spin-Echo T1-Weighted Black-Blood Vessel Wall Imaging: A Preliminary Study

**DOI:** 10.3390/diagnostics12041014

**Published:** 2022-04-17

**Authors:** Katsuhiro Inoue, Ryohei Nakayama, Shiho Isoshima, Shinichi Takase, Tsunehiro Yamahata, Maki Umino, Masayuki Maeda, Hajime Sakuma

**Affiliations:** 1Department of Radiology, Mie University Hospital, Tsu 514-8507, Mie, Japan; inokatsu@med.mie-u.ac.jp (K.I.); shiho-i@med.mie-u.ac.jp (S.I.); tkss1902@med.mie-u.ac.jp (S.T.); ty373@med.mie-u.ac.jp (T.Y.); 2Department of Electronic and Computer Engineering, Ritsumeikan University, Kusatsu 525-8577, Shiga, Japan; nakayama@med.mie-u.ac.jp; 3Department of Radiology, Mie University School of Medicine, Tsu 514-8507, Mie, Japan; m-tochio@med.mie-u.ac.jp (M.U.); sakuma@med.mie-u.ac.jp (H.S.); 4Department of Neuroradiology, Mie University School of Medicine, Tsu 514-8507, Mie, Japan

**Keywords:** carotid artery, unstable carotid plaque, semiautomated segmentation, 3D TSE T1-weighted black-blood vessel wall imaging

## Abstract

Unstable carotid plaques are visualized as high-signal plaques (HSPs) on 3D turbo spin-echo T1-weighted black-blood vessel wall imaging (3D TSE T1-BB VWI). The purpose of this study was to compare manual segmentation and semiautomated segmentation for the quantification of carotid HSPs using 3D TSE T1-BB VWI. Twenty cervical carotid plaque lesions in 19 patients with a plaque contrast ratio of > 1.3 compared to adjacent muscle were studied. Using the mean voxel value for the adjacent muscle multiplied by 1.3 as a threshold value, the semiautomated software exclusively segmented and measured the HSP volume. Manual and semiautomated HSP volumes were well correlated (r = 0.965). Regarding reproducibility, the inter-rater ICC was 0.959 (bias: 24.63, 95% limit of agreement: −96.07, 146.35) for the manual method and 0.998 (bias: 15.2, 95% limit of agreement: −17.83, 48.23) for the semiautomated method, indicating improved reproducibility by the semiautomated method compared to the manual method. The time required for semiautomated segmentation was significantly shorter than that of manual segmentation times (81.7 ± 7.8 s versus 189.5 ± 49.6 s; *p* < 0.01). The results obtained in this study demonstrate that the semiautomated segmentation method allows for reliable assessment of the HSP volume in patients with carotid plaque lesions, with reduced time and effort for the analysis.

## 1. Introduction

The presence of intraplaque hemorrhage (IPH) in cervical carotid arteries is a strong predictor of future ischemic events [1,2]. T1-weighted black-blood (T1-BB) vessel wall imaging (VWI) with magnetization-prepared 3D gradient echo (GRE) sequence was initially introduced as a method that can provide high capacity for the detection and quantification of IPH [1,2,3]. IPH shows a high-signal intensity due to T1 shortening caused by methemoglobin. Recently, 3D turbo spin-echo (TSE) T1-BB VWI is becoming more widely used for the evaluation of the carotid artery and intracranial artery lesions [4,5,6]. This method exhibits fewer flow artifacts for delineating high-signal plaques (HSPs) in carotid artery stenosis compared with 3D GRE T1-BB VWI [7]. Unlike 3D GRE T1-BB VWI, 3D TSE T1-BB VWI can differentiate among the three plaque types, including IPH, lipid-rich/necrotic, and fibrous plaques, using the contrast ratios of the plaques compared to the adjacent muscles, with the cutoff values of 1.52 (IPH vs. lipid-rich/necrotic) and 1.3 (lipid-rich/necrotic vs. fibrous) in the literature [8]. Therefore, unstable plaques, such as hemorrhagic and/or lipid-rich/necrotic plaques, can be delineated as lesions with HSP > 1.3 on 3D TSE T1-BB VWI.

Manually measuring plaque volume is time-consuming and may incur bias due to different levels of reviewer experience. A semiautomated approach to carotid plaque segmentation and quantification may reduce both measurement variability and manual effort. Several recent studies have demonstrated that semiautomated segmentation and volume measurements of IPH are feasible for 3D GRE T1-BB VWI [9,10,11,12]. However, these studies only focused on IPH volume, not the volume of unstable plaques, such as lipid-rich/necrotic plaques without hemorrhage, which is another type of unstable plaque component.

We developed semiautomated software for HSP segmentation and volume measurement. We hypothesized that unstable carotid plaque volume could be efficiently quantified by analyzing 3D TSE T1-BB VWI with the semiautomated approach. Our aim was to compare manual segmentation and semiautomated segmentation for the quantification of carotid HSPs using 3D TSE T1-BB VWI and to evaluate the reproducibility of these carotid HSP volume measurements.

## 2. Materials and Methods

### 2.1. Patients

This retrospective study was approved by the institutional review board, which waived the requirement for informed consent. Patient anonymity was ensured prior to the assessment of the data. Seventy-five patients with carotid stenosis who underwent MRI examination were recruited for this study. Then, the patients who met the following inclusion criteria constituted the study subjects; (1) had atherosclerotic plaques in at least one carotid artery identified by ultrasound imaging; (2) had sufficient MR image quality of carotid lesions; (3) had the HSP with a contrast ratio of > 1.3 compared to adjacent muscle in entire plaques on 3D TSE T1-BB VWI. One reviewer (K.I.) manually measured the contrast ratios for the entire carotid plaques in each case. Eventually, 19 patients (age range, 65–91 years [mean, 76.8 years]; 19 men) with 20 carotid lesions were qualified for this study.

### 2.2. MRI Protocol

Imaging was performed using a 3T MRI (Ingenia, Philips Healthcare, Best, The Netherlands) with a dS Head-Neck Spine coil. The imaging-sequence parameters of 3D TSE T1-BB VWI were: T1-VISTA (Volume ISotropic Tse Acquisition), TR/TE = 350 ms/23 ms, refocusing angle = 40°, TSE factor = 11, in-plane spatial resolution = 0.90 mm × 0.90 mm, reconstructed resolution = 0.45 mm × 0.45 mm, slice thickness = 0.9 mm, number of excitations = 1, and acquisition time = 4 min 23 s. 

### 2.3. Segmentation of HSP by Software

Custom software for semiautomated segmentation of HSP was developed based on MATLAB 2019b. The software consists of two sequential processes (Figure 1): (1) extraction of the plaque region with a region growing process and (2) segmentation of the HSP region from the extracted plaque region based on the mean voxel value of the adjacent muscle.

For semiautomated segmentation, the seed points were set manually near the plaque center in the 3D TSE T1-BB VWI images. For reproducible measurements, the voxel with the highest voxel value in a 7 × 7 × 7 region centered on the manual seed point was defined as the corrected seed point. To reduce the influence of noise, 3D TSE T1-BB VWI images were smoothed using a Gaussian filter with σ = 0.5. The region growing method was applied to the smoothed images to extract the plaque region. In the region growing process, the initial threshold value was given by the highest voxel value in the 7 × 7 × 7 region centered on the corrected seed point. Neighboring voxels to the corrected seed point with voxel values higher than the threshold value were added to the plaque region. This process was repeated while reducing the threshold value by one until one of the following conditions was satisfied: (i) no new voxels were added to the plaque region and (ii) the volume of the plaque region suddenly more than doubled.

For segmentation of the HSP region, the region of interest (ROI) for muscle tissue was manually set near the plaque. The mean voxel value in the ROI for muscle tissue multiplied by a coefficient of 1.3 was used as a threshold value for segmenting the HSP region from the extracted plaque region [8].

### 2.4. Image Analyses

HSPs were independently segmented by two readers (K.I. and S.I.) with > 5 years of experience in MR imaging using the manual method and the semiautomated software. Image analyses included two steps. First, using the mean voxel value for the adjacent muscle multiplied by 1.3 as the threshold value, the semiautomated software exclusively segmented HSP plaques > 1.3 of the contrast ratios of the plaques to adjacent muscles, and plaque volumes were measured. Readers were also asked to manually segment the HSP. HSP volume was calculated as the sum of the plaque area within slices multiplied by slice thickness. The manual processes were performed on the EVInsite R (public and social systems solution provider, Tokyo, Japan). Segmentation time was measured and averaged for the two readers both in the manual and semiautomated methods. Then, we compared the differences between the manual review and the semiautomated review for the segmentation of HSP.

### 2.5. Statistical Analyses of Data

Intraclass correlation coefficients (ICC) were used to assess the reproducibility of the two methods. Both the inter-rater and intra-rater agreements for the segmentation of carotid HSP were assessed. HSP was segmented by one reader twice with a time interval of two weeks to minimize the memory bias. HSP was independently segmented by another reader for testing the inter-rater agreement.

Validity was evaluated using Spearman’s Rank Correlation and Bland–Altman analyses. In addition, segmentation time was assessed for the manual method and the semiautomated method. Based on 95% confidence intervals of the ICC estimate, values less than 0.5, between 0.5 and 0.75, between 0.75 and 0.9, and greater than 0.90 were indicative of poor, moderate, good, and excellent reliability, respectively [13]. A *p*-value less than 0.05 was considered statistically significant. Statistical analyses were performed using SPSS 26.0 (IBM, Chicago, IL, USA).

## 3. Results

HSP volumes measured by the manual and semiautomated methods were correlated well (r = 0.965, Figure 2). Bland–Altman analysis showed that the bias between manual and semiautomated methods was very small (bias: −15.53, Figure 3). Regarding reproducibility, the inter-rater ICC was 0.959 (bias: 24.63, 95% limit of agreement: −96.07, 146.35) and the intra-rater ICC was 0.979 (bias: 3.29, 95% limit of agreement: −99.91, 106.51) for the manual method (Figure 4a,b). For the semiautomated method, the inter-rater ICC was 0.998 (bias: 15.2, 95% limit of agreement: −17.83, 48.23) and the intra-rater ICC was 0.998 (bias: 0.95, 95% limit of agreement: −26.97, 28.87) (Figure 5a,b). The narrower limit of agreements by the semiautomated method suggested a better reproducibility compared to the manual method.

The time required for segmentation was significantly reduced with the semiautomated method compared with the manual method (81.7 ± 7.8 s versus 189.5 ± 49.6 s; *p* < 0.01, Figure 6). The segmentation times using the semiautomated method were relatively constant regardless of the HSP volumes, ranging from 93 to 118 s. In contrast, the manual method exhibited a wide range of segmentation times (124 to 307 s) with a linear relationship between HSP volumes and segmentation time (*p* < 0.01, Figure 7).

## 4. Discussion

In the current study, we developed a semiautomated method for HSP segmentation and quantification in 3D TSE T1-BB VWI. The HSP volume determined by the semiautomated method showed an excellent agreement with the manual method in 20 carotid plaque lesions. In addition, the assessments of inter-rater and intra-rater reproducibility revealed that the semiautomated method had substantially narrower 95% confidence intervals compared to the manual assessment. The results obtained in this study indicated that the semiautomated segmentation method allows for reliable assessment of the HSP volume in patients with carotid plaque lesions, with reduced time and effort for the analysis.

For the assessment of carotid plaque vulnerability by MR imaging, 3D GRE T1-BB VWI was initially employed to detect IPH in the carotid lesions [1,2,3]. Recently, 3D TSE T1-BB VWI has been widely used for the evaluation of vessel wall lesions due to its advantages, including less flow artifacts in the vessel lesion [7] and its applicability to intracranial vessels [5]. Regarding the evaluation of unstable plaques, 3D TSE T1-BB VWI can discriminate lipid-rich/necrotic plaques without hemorrhage from IPH, while 3D GRE T1-BB VWI does not [8]. Consequently, in this study, 3D TSE T1-BB VWI was used as an imaging technique to evaluate unstable plaques.

In this study, we evaluated the reproducibility of the developed software in comparison to the manual method as the benchmark. Although reproducibility was excellent in both manual and semiautomated methods, 95% limit of agreements of inter-rater and intra-rater were significantly narrower in the semiautomated method than in the manual method, indicating that the semiautomated method is better in reproducibility. In several multicenter studies, plaque lesions, such as IPH, were manually segmented [14,15,16,17]. The semiautomated method would be useful in multicenter studies because it can reduce segmentation bias among raters. Our results showed that the semiautomated method significantly reduced the time required for segmentation compared to the manual method. The shorter segmentation time is advantageous in the clinical setting because 3D TSE T1-BB VWI can provide high-resolution images with wider anatomical coverage, particularly for carotid plaque lesions with extremely large diameters and lengths [4,6,7]. The semiautomated method developed in this study would reduce the effort required to analyze a large 3D image dataset and may improve the confidence interval of the measurements.

Vulnerable plaques include lipid-rich/necrotic core and IPH. The lipid-rich/necrotic core is a collection of heterogeneous materials within the atherosclerotic plaques that consist of crystals and necrotic debris of apoptotic cells [18]. Lipid-rich/necrotic core increases the risk to rupture when its size increases [19]. However, CT is difficult to differentiate lipid-rich/necrotic core from IPH due to similar CT attenuation [18]. It was reported that the presence of IPH stimulates the progression of carotid atherosclerotic plaques [20] and that it is also responsible for increased stroke incidence and stroke recurrence [21]. IPH is a risk for emboli in the procedures of carotid artery stenting [9] or in intravascular ultrasound for the evaluation of carotid artery atherosclerotic lesions [22]. Ultrasound and CT are less suitable methods for the detection of IPH than MRI [18]. Therefore, we need to be cautious about the diagnosis of vulnerable plaques.

Most of the previous studies focused on the presence or absence of IPH [1,2,3,4,6] or IPH volume [3,9,14,15,16]. Although the 3D GRE T1-BB VWI approach showed high diagnostic capability for the detection and quantification of IPH [3], the ability to diagnose lipid-rich/necrotic plaque without hemorrhage is generally limited. However, the lipid-rich/necrotic plaque without hemorrhage is another type of vulnerable plaque that requires more attention. This is the first report on semiautomated quantification of unstable plaque volumes by using 3D TSE T1-BB VWI.

The current study has a few limitations. First, the study group included a relatively small number of cases. In addition, no histological confirmation was performed because our study did not include patients who underwent carotid endarterectomy to histologically validate the carotid plaque components. Although our study was based on the data by Narumi et al. [8], which used 3D TSE T1-BB VWI with histologic validation, we need to investigate histologic-radiologic comparison with a larger number of cases in a future study. Second, the cutoff value of 1.52 was reported to discriminate IPH from lipid-rich/necrotic plaque without hemorrhage [8]. The software used in this study may exclusively segment IPH using the mean voxel value for the adjacent muscle multiplied by 1.52 as the threshold value. However, we did not investigate this issue in the current study. Further study is needed to clarify this point in the future.

## 5. Conclusions

We developed semiautomated software that allows for the quantification of HSP volume in carotid plaques. Although we need to investigate histologic-radiologic comparison with a larger number of cases in a future study, the HSP volumes determined by the new approach was validated with those by manual assessments in 20 carotid plaque lesions. In addition to the excellent agreement in the HSP volume by two methods, improved inter-rater and intra-rater reproducibility by the semiautomated method was indicated by smaller 95% confidence intervals compared to the manual assessment. The semiautomated segmentation method allows for reliable assessment of the HSP volume in patients with carotid plaque lesions, with reduced time and effort for the analysis.

## Figures and Tables

**Figure 1 diagnostics-12-01014-f001:**
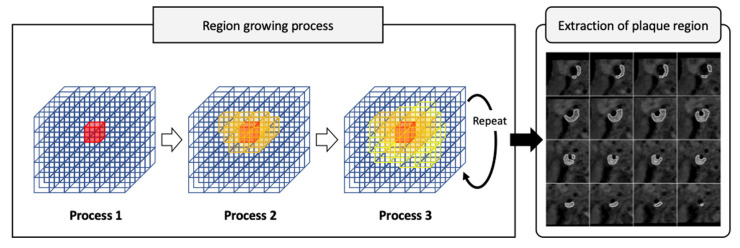
Flow chart of semiautomated segmentation of high-signal plaque (HSP) using custom-designed software. Region growing process consists of the 3 steps. In process 1, initial threshold value is given by the highest voxel value in the 7 × 7 × 7 region centered on the seed point. In process 2, neighboring voxels with voxel values higher than the threshold value are added to the plaque region. In process 3, addition to plaque region is repeated while reducing the threshold value by one until one of the end conditions is satisfied. Then, extraction of plaque region is achieved on 3D TSE T1-BB VWI images.

**Figure 2 diagnostics-12-01014-f002:**
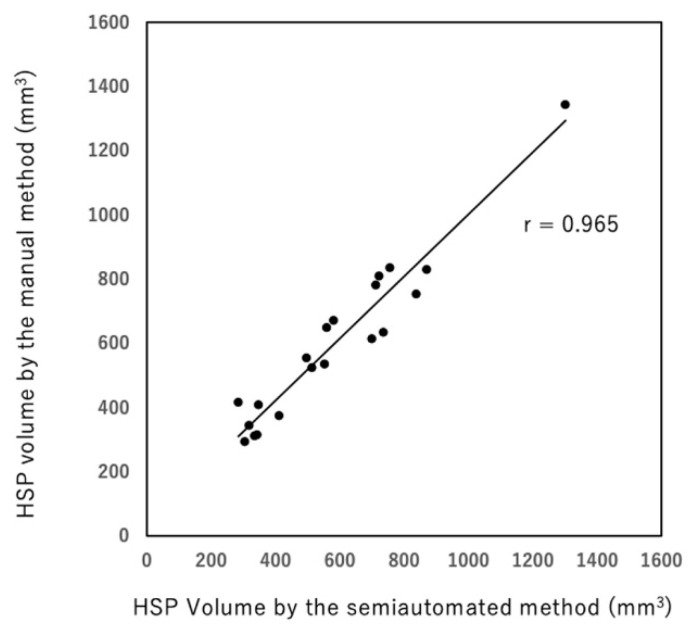
Correlation of HSP volumes measured by the manual and semiautomated segmentation methods (r = 0.965).

**Figure 3 diagnostics-12-01014-f003:**
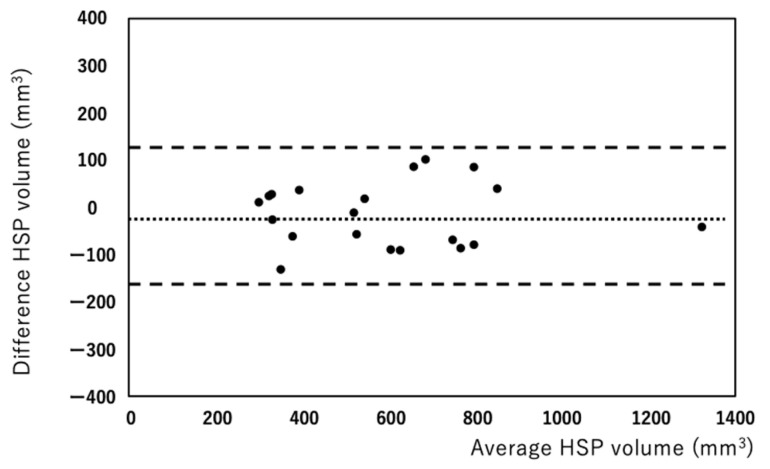
Bland–Altman analysis shows bias between the manual and semiautomated methods is very small (bias: −15.5 mm^3^).

**Figure 4 diagnostics-12-01014-f004:**
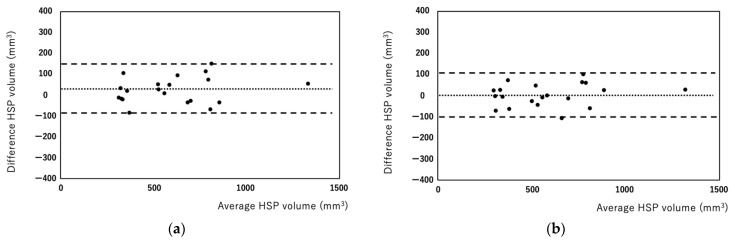
Bland–Altman plots of inter-rater (**a**) and intra-rater (**b**) HSP volumes measured using the manual method. Each plot represents the percentage difference between the measurements by the two readers (**a**) and between the measurements of the two reading sessions by a single rater (**b**). The intraclass correlation coefficients were 0.959 for inter-raters (**a**) and 0.979 for intra-raters (**b**).

**Figure 5 diagnostics-12-01014-f005:**
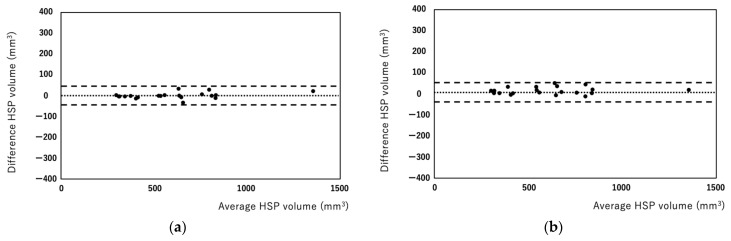
Bland–Altman plots of inter-rater (**a**) and intra-rater (**b**) HSP volume measured using the semiautomated method. Each plot represents the percentage difference between the measurements by the two readers (**a**) between the measurements of the two reading sessions by a single rater (**b**). The intraclass correlation coefficients were 0.998 for inter-raters (**a**) and 0.998 for intra-raters (**b**).

**Figure 6 diagnostics-12-01014-f006:**
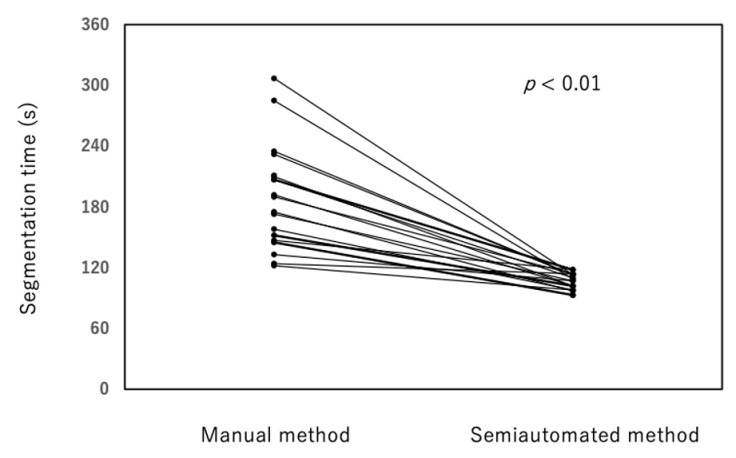
The segmentation times using the semiautomated method were shorter than segmentation times using the manual method (81.7 ± 7.8 s versus 189.5 ± 49.6 s; *p* < 0.01).

**Figure 7 diagnostics-12-01014-f007:**
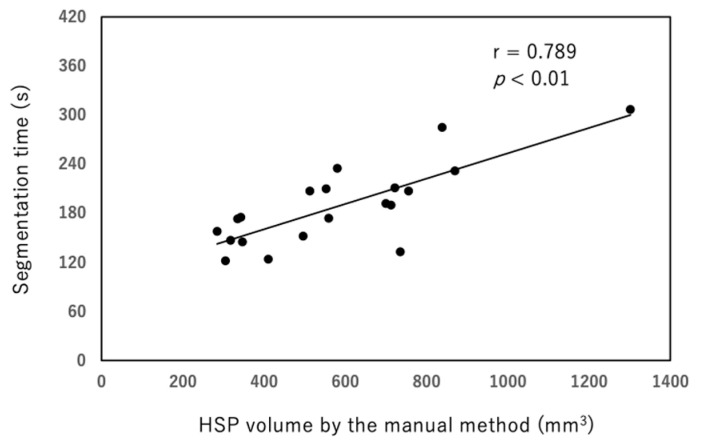
The manual method exhibited a wide range of segmentation times (124 to 307 s) and a linear relationship between HSP volume and segmentation times (*p* < 0.01, r = 0.789).

## Data Availability

Not applicable.

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
