# Peer review of "Semiautomated Segmentation and Volume Measurements of Cervical Carotid High-Signal Plaques Using 3D Turbo Spin-Echo T1-Weighted Black-Blood Vessel Wall Imaging: A Preliminary Study"

_diagnostics, 2022, doi:10.3390/diagnostics12041014_

Round 1
Reviewer 1 Report
This is a well presented MRI study of the vulnerable carotid plaque. The study novelty comprise a non-invasive assessment of high-risk carotid plaques with new MRI technique. Authors fond that IPH volume measurements on the new MRI technique are highly reproducible using semiautomatic measurement. This is an important finding.
Introduction, methods and study results are well presented. However, the study group is relatively small, and it lacks histopatological confirmation. Thus, the study limitation section should be improved, and conclusions should be written with a greater caution, as the new technique needs to be confirmed by the ongoing studies.
I think that a paragraph in discussion on the importance of the detection of the intra-plaque hemorrhage within the plaque should be added. It was proven that the presence of intraplaque hemorrhage stimulates progression of carotid atherosclerotic plaques at 18 months follow-up (Takaya, N., Yuan, C., Chu, B., Saam, T., Polissar, N. L., Jarvik, G. P., Isaac, C., McDonough, J., Natiello, C., Small, R., Ferguson, M. S., & Hatsukami, T. S. (2005). Presence of intraplaque hemorrhage stimulates progression of carotid atherosclerotic plaques: a high-resolution magnetic resonance imaging study. Circulation, 111(21), 2768–2775. https://doi.org/10.1161/CIRCULATIONAHA.104.504167)
It is also responsible for increased stroke incidence and stroke recurrence (van Veelen, A.; van der Sangen, N.M.R.; Delewi, R.; Beijk, M.A.M.; Henriques, J.P.S.; Claessen, B.E.P.M. Detection of Vulnerable Coronary Plaques Using Invasive and Non-Invasive Imaging Modalities. J. Clin. Med. 2022, 11, 1361. https://doi.org/10.3390/jcm11051361 , and Zaman, R. T., Kosuge, H., Gambhir, S. S., & Xing, L. (2021). Detection of Carotid Artery Stenosis with Intraplaque Hemorrhage and Neovascularization Using a Scanning Interferometer. Nano letters, 21(13), 5714–5721.
And also, a comment on the limitations of the other imaging tools to differentiate the lipid-core from fibrous plaque, and from a hemorrhagic plaque, such as intravascular ultrasonography (IVUS), computed tomography, optical coherence tomography, and scanning interferometer would be appreciated. In addition some procedures can be based on the direct plaque assessment, and IVUS-triggered carotid stenting procedures can prevent cerebral embolization. (e.g. Musialek, P., Pieniazek, P., Tracz, W., Tekieli, L., Przewlocki, T., Kablak-Ziembicka, A., Motyl, R., Moczulski, Z., Stepniewski, J., Trystula, M., Zajdel, W., Roslawiecka, A., Zmudka, K., & Podolec, P. (2012). Safety of embolic protection device-assisted and unprotected intravascular ultrasound in evaluating carotid artery atherosclerotic lesions. Medical science monitor : international medical journal of experimental and clinical research, 18(2), MT7–MT18. https://doi.org/10.12659/msm.882452)
This would emphesize why novel MRI techniques are needed, and why we care so much about plaque hemorrahe or neovascularisation.
Author Response
This is a well presented MRI study of the vulnerable carotid plaque. The study novelty comprise a non-invasive assessment of high-risk carotid plaques with new MRI technique. Authors fond that IPH volume measurements on the new MRI technique are highly reproducible using semiautomatic measurement. This is an important finding.
Introduction, methods and study results are well presented. However, the study group is relatively small, and it lacks histopatological confirmation. Thus, the study limitation section should be improved, and conclusions should be written with a greater caution, as the new technique needs to be confirmed by the ongoing studies.
According to the reviewer’s comment, we revised the limitation section as follows; First, the study group is relatively small number of cases. In addition, no histological confirmation was performed because our study did not include patients who underwent carotid endarterectomy to histologically validate the carotid plaque components. Although our study was based on the data by Narumi et al. [8], which used 3D TSE T1-BB VWI with histologic validation, we need to investigate histologic-radiologic comparison with larger number of cases in a future study.
We revised the conclusion as follows; Although we need to investigate histologic-radiologic comparison with larger number of cases in a future study, the HSP volumes determined by the new approach was validated with those by manual assessments in 20 carotid plaque lesions.
I think that a paragraph in discussion on the importance of the detection of the intra-plaque hemorrhage within the plaque should be added. It was proven that the presence of intraplaque hemorrhage stimulates progression of carotid atherosclerotic plaques at 18 months follow-up (Takaya, N., Yuan, C., Chu, B., Saam, T., Polissar, N. L., Jarvik, G. P., Isaac, C., McDonough, J., Natiello, C., Small, R., Ferguson, M. S., & Hatsukami, T. S. (2005). Presence of intraplaque hemorrhage stimulates progression of carotid atherosclerotic plaques: a high-resolution magnetic resonance imaging study. Circulation, 111(21), 2768–2775. https://doi.org/10.1161/CIRCULATIONAHA.104.504167)
It is also responsible for increased stroke incidence and stroke recurrence (van Veelen, A.; van der Sangen, N.M.R.; Delewi, R.; Beijk, M.A.M.; Henriques, J.P.S.; Claessen, B.E.P.M. Detection of Vulnerable Coronary Plaques Using Invasive and Non-Invasive Imaging Modalities. J. Clin. Med. 2022, 11, 1361. https://doi.org/10.3390/jcm11051361 , and Zaman, R. T., Kosuge, H., Gambhir, S. S., & Xing, L. (2021). Detection of Carotid Artery Stenosis with Intraplaque Hemorrhage and Neovascularization Using a Scanning Interferometer. Nano letters, 21(13), 5714–5721.
And also, a comment on the limitations of the other imaging tools to differentiate the lipid-core from fibrous plaque, and from a hemorrhagic plaque, such as intravascular ultrasonography (IVUS), computed tomography, optical coherence tomography, and scanning interferometer would be appreciated. In addition, some procedures can be based on the direct plaque assessment, and IVUS-triggered carotid stenting procedures can prevent cerebral embolization. (e.g. Musialek, P., Pieniazek, P., Tracz, W., Tekieli, L., Przewlocki, T., Kablak-Ziembicka, A., Motyl, R., Moczulski, Z., Stepniewski, J., Trystula, M., Zajdel, W., Roslawiecka, A., Zmudka, K., & Podolec, P. (2012). Safety of embolic protection device-assisted and unprotected intravascular ultrasound in evaluating carotid artery atherosclerotic lesions. Medical science monitor : international medical journal of experimental and clinical research, 18(2), MT7–MT18. https://doi.org/10.12659/msm.882452)
This would emphesize why novel MRI techniques are needed, and why we care so much about plaque hemorrahe or neovascularisation.
According to the reviewer’s comments, we added a paragraph in the discussion as follows; Vulnerable plaques include lipid-rich/necrotic core and IPH. lipid-rich/necrotic core is a collection of heterogeneous materials within the atherosclerotic plaques that consist of crystals and necrotic debris of apoptotic cells [18]. Lipid-rich/necrotic core increases its risk to rupture when its size is increasing [19]. However, CT is difficult to differentiate lipid-rich/necrotic core from IPH due to similar CT attenuation [18]. It was reported that the presence of IPH stimulates progression of carotid atherosclerotic plaques [20] and that it is also responsible for increased stroke incidence and stroke recurrence [21]. IPH is a risk for emboli in the procedures of carotid artery stenting [9] or in intravascular ultrasound for the evaluation of carotid artery atherosclerotic lesions [22]. Ultrasound and CT are less suitable methods for detection of IPH than MRI [18]. Therefore, we need to be cautious about diagnosis of vulnerable plaques.
- Saba L, Saam T, JägerHR, Yuan C, Hatsukami TS, Saloner D, Wasserman BA, Bonati LH, Wintermark M. Imaging biomarkers of vulnerable carotid plaques for stroke risk prediction and their potential clinical implications. Lancet Neurol 2019, 18, 559-572.
- Ota H, Yu W, Underhill HR, Oikawa M, Dong L, Zhao X, Polissar NL, Neradilek B, Gao T, Zhang Z, Yan Z, Guo M, Zhang Z, Hatsukami TS, Yuan C. Hemorrhage and large lipid-rich necrotic cores are independently associated with thin or ruptured fibrous caps: an in vivo 3T MRI study. Atherioscler Thromb Vasc Biol 2009, 29, 1696-1701.
- Takaya N, Yuan C, Chu B, Saam T, Polissar NL, Jarvik GP, Isaac C, McDonough J, Natiello C, Small R, Ferguson MS, Hatsukami TS. Presence of intraplaque hemorrhage stimulates progression of carotid atherosclerotic plaques: a high-resolution magnetic resonance imaging study. Circulation 2005, 111, 2768–2775.
- Schindler A, Schinner R, Altaf N, Hosseini AA, Simpson RJ, Esposito-Bauer L, Singh N, Kwee RM, Kurosaki Y, Yamagata S, Yoshida K, Miyamoto S, Maggisano R, Moody AR, Poppert H, Kooi ME, Auer DP, Bonati LH, Saam T. Prediction of stroke risk by detection of hemorrhage in carotid plaques: meta-analysis of individual patient data. JACC Cardiovasc Imaging 2020, 13, 395-406.
- Musialek P, Pieniazek P, Tracz W, Tekieli L, Przewlocki T, Kablak-Ziembicka A, Motyl R, Moczulski Z, Stepniewski J, Trystula M, Zajdel W, Roslawiecka A, Zmudka K, Podolec P. Safety of embolic protection device-assisted and unprotected intravascular ultrasound in evaluating carotid artery atherosclerotic lesions. Med Sci Monit 2012, 18, MT7-18.

Reviewer 2 Report
I read with great attention and interest the paper entitled "Semiautomated Segmentation and Volume Measurements of 2 Cervical Carotid High-Signal Plaques using 3D Turbo Spin-3 Echo T1-Weighted Black-Blood Vessel Wall Imaging"
Authors reported a well-designed and interesting preliminary study on semi-automated software to evaluate vulnerable carotid plaques.
I have some minor suggestion for the Authors:
- better underline in the title ant whole manuscript that the present is a pilot/feasibility/pivotal studi, because you have not enough patients to be conclusive
- reduce the relative importance given to the time sparing with the new software, although statistically significant a difference of one minute and half is not significant in real life
Author Response
I read with great attention and interest the paper entitled "Semiautomated Segmentation and Volume Measurements of 2 Cervical Carotid High-Signal Plaques using 3D Turbo Spin-3 Echo T1-Weighted Black-Blood Vessel Wall Imaging"
Authors reported a well-designed and interesting preliminary study on semi-automated software to evaluate vulnerable carotid plaques.
I have some minor suggestion for the Authors:
- better underline in the title ant whole manuscript that the present is a pilot/feasibility/pivotal studi, because you have not enough patients to be conclusive
According to reviewer’ s comment, we revised the title as follows; “Semiautomated Segmentation and Volume Measurements of Cervical Carotid High-Signal Plaques using 3D Turbo Spin-Echo T1-Weighted Black-Blood Vessel Wall Imaging: A Preliminary Study”
In limitation, we revised as follows; First, the study group is relatively small number of cases. In addition, no histological confirmation was performed because our study did not include patients who underwent carotid endarterectomy to histologically validate the carotid plaque components. Although our study was based on the data by Narumi et al. [8], which used 3D TSE T1-BB VWI with histologic validation, we need to investigate histologic-radiologic comparison with larger number of cases in a future study.
In conclusion parts, we revised as follows; Although we need to investigate histologic-radiologic comparison with larger number of cases in a future study, the HSP volumes determined by the new approach was validated with those by manual assessments in 20 carotid plaque lesions.
- reduce the relative importance given to the time sparing with the new software, although statistically significant a difference of one minute and half is not significant in real life
I understand what you are saying but reducing pain and segmentation times are awarded package by using this software. In a reality, we asked reviewers to perform manual segmentation as fast as they can in this study and they did it because they had to, but any reviewer was feeling painful doing that. Manual segmentation needs intense concentration, leading to pain on any reviewer. I could not quantify the degree of pain burden on reviewers, so in this study we just quantified the segmentation time.

Round 2
Reviewer 1 Report
I do appreciate changes made by the Authors. Especially, I do like a richer discussion, now well positioning Author's findings in the light of the other imaging tools.